# Fair Reinforcement Learning for Just AI

**Ezgi Korkmaz**

## Abstract

Currently the most powerful AI systems are aligned with human values via reinforcement learning from human feedback. Yet, reinforcement learning from human feedback models human preferences as noisy samples from a single linear ordering of shared human values and is unable to incorporate democratic AI alignment. In particular, the standard approach fails to represent and reflect diverse and conflicting perspectives of human values. Recent research introduced the theoretically principled notion of quantile fairness for training a reinforcement learning policy in the presence of multiple, competing sets of values from different agents. Quite recent work provided an algorithm for achieving quantile fairness in the tabular setting with explicit access to the full set of states, actions and transition probabilities in the MDP. These current methods require solving linear programs with the size of the constraint set given by the number of states and actions, making it unclear how to translate this into practical training algorithms that can only take actions and observe individual transitions from the current state. In this paper, we design and prove the correctness of a new algorithm for quantile fairness that makes efficient use of standard policy optimization as a black-box without any direct dependence on the number of states or actions. We further empirically validate our theoretical results and demonstrate that our algorithm achieves competitive fairness guarantees to the prior work, while being orders of magnitude more efficient with respect to computation and the required number of samples. Our algorithm opens a new avenue for provable fairness guarantees in any setting where standard policy optimization is possible.

## 1 Introduction

Reinforcement learning is at the core of the most powerful AI agents from scientific and mathematical reasoning to alignment with human values. Standard reinforcement learning reduces collective welfare to the maximization of a single scalar reward, assuming the existence of a monolithic reward structure that is fundamentally incapable of representing a multiplicity of diverse and conflicting objectives. While modern AI alignment relies on optimizing toward a consensus objective, such models fail to capture heterogeneous and competing human values. As a result there has been growing recognition of the need to develop *pluralistic alignment* methods (Conitzer et al., 2024; Sorensen et al., 2024), that align AI systems by taking into account the existence of multiple different reward or utility functions representing diverse human preferences. More generally, as powerful AI systems are further integrated into society, it is crucial to develop algorithms that aggregate diverse, competing objectives and human preferences in a rigorous way.

The idea of *policy aggregation* presents an initial theoretical approach to the problem of value alignment from the preferences of multiple agents (Alamdari et al., 2024). The main approach is to apply rigorous methods from computational social choice theory to reinforcement learning with multiple diverse reward functions. However, computational social choice typically deals with preference rankings over discrete sets of alternatives, which cannot be applied to the continuous space of possible stochastic policies in reinforcement learning. The key insight of Alamdari et al. (2024) is that one can still compute a continuous analogue of a "ranking" for each agent, by assigning to each policy a score of $q$ if the agent receives a higher score than a $q$-fraction of all possible alternative policies. Critically, the score $q$ only depends on the relative ranking of policies in the MDP, and thus provides a theoretically principled way to compare preferences across different agents that may have very different reward scales.

The prior approach to policy aggregation examines multiple approaches to designing mechanisms that produce a fair policy given the notion of ranking described above. The authors show that there are polynomial time algorithms to compute such fairness mechanisms, under the assumption that a full, explicit description of the MDP is available. That is, the mechanism needs to have access to the table of transition probabilities in the MDP, and runs in time polynomial in the number of states and actions. Such methods are clearly intractable in the deep learning setting. To design algorithms that can be directly adapted to high dimensional environments, one typically assumes access to a *policy optimization oracle* i.e. a subroutine that can optimize a policy in a given MDP. Then, the goal becomes to design a provably *oracle-efficient algorithm*, meaning that the algorithm makes a polynomial number of calls to the policy optimization oracle, and has no other explicit access to the MDP in question. Therefore, in this paper, we study the following question:

*Are there provably oracle-efficient algorithms for fair reinforcement learning?*

Our main results provide a positive answer to the above question.

**Our Results.** The notion of fairness that we consider is that of *quantile fairness* introduced by Babichenko et al. (2024). This notion states that a policy is $q$-quantile fair, if it provides return in the top $q$-th quantile for each agent simultaneously. Alamdari et al. (2024) provides a provably efficient algorithm to achieve $q$-quantile fairness for the maximum possible value of $q$, when given explicit access to the full table of transition probabilities of the MDP. Our main result is an oracle-efficient algorithm for $q$-quantile fair reinforcement learning.

**Theorem 1.1** (Informal)**.** *Given an MDP with $n$ agents, each with a different reward function, there is an algorithm that makes $O(n)$ calls to a policy optimization oracle, and returns a $q$-quantile fair policy for the maximum possible value of $q$.*

**Technical Overview.** To obtain this result, we first have to address a drawback of the initial definition of $q$-quantile fairness. The quantile $q$ is defined by comparing a given policy's return to that of all possible alternative policies in the MDP. Algorithms for $q$-quantile fairness thus naturally rely on the ability to estimate quantiles. This is achieved by Alamdari et al. (2024) by sampling from a natural choice of uniform distribution on the set of all possible policies, conditioned on achieving value in the $q$-th quantile for a given reward function. While such sampling can be done in polynomial time given explicit access to the full table of transition probabilities of the MDP, it is intractable in general. In particular, we show in Section 5 that there are MDPs where the fraction of policies with non-zero reward is exponentially small in the number of states of the MDP. This result implies that uniform sampling to estimate the quantile function has exponential sample complexity. To circumvent this issue, we instead consider $q$-quantile fairness with respect to a different distribution on policies, induced by individually optimal policies for each agent. We then prove in Section 6.1 that, using $O(n)$ evaluations of a policy optimization oracle, it is possible to accurately estimate the quantile function for this distribution.

Next, given the ability to estimate quantiles, the algorithm of Alamdari et al. (2024) utilizes a linear program defined over the states and actions of the MDP in order to compute a $q$-quantile fair policy. In Section 6.2, we instead design a new algorithm that utilizes the multiplicative weights update method to achieve the same goal, while only requiring $O(\log n)$ calls to a policy evaluation oracle.

## 2 RELATED WORK

Most closely related to our work, the notion of fair policy aggregation was introduced by Alamdari et al. (2024), which gave provably efficient algorithms in the setting where the transition function of the MDP is given explicitly as a table of transition probabilities. Babichenko et al. (2024) defined the notion of quantile fairness in computational social choice theory.

Due to the use of reinforcement learning from human feedback (Christiano et al., 2017; Bai et al., 2022) for aligning the most advanced AI systems, notions of fairness in reinforcement learning with multiple reward functions are closely related to the application of social choice theory to AI alignment (Conitzer et al., 2024). Recent research in this direction has discovered that standard RLHF based on the Bradley-Terry preference model (Bradley & Terry, 1952) implicitly aggregates reward functions via the Borda count social-choice rule. Concurrent work has designed RLHF methods

based on computing a *von Neumann winner* policy (Swamy et al., 2024; Wang et al., 2023), which is equivalent to the rule from social choice theory known as *maximal lotteries* (Freund & Schapire, 1997; Kreweras, 1965).

Early work on multi-objective reinforcement learning While (1982) developed algorithms that find pareto-efficient policies. In the decades since, there has been extensive work on fairness in multi-objective RL, with a significant focus on methods that combine multiple reward functions into a single reward, which is then optimized Reuel & Ma (2024). The work of Zhang & Shah (2014) uses a regularized maximin objective, which attempts to balance the sum of the rewards against the maximization of the minimum reward. Ogryczak et al. (2013) study the Gini social welfare as its reward combination method. More generally, there has been extensive work focused on various other methods that numerically combine multiple reward functions into a single one e.g. the Gini coefficient or the Nash welfare Ju et al. (2024); Hayes et al. (2022); Siddique et al. (2020). These methods inherently assume that rewards can be normalized and compared across agents. However, when inferring rewards from agent behavior, it is only possible to learn the rewards up to a positive affine transformation. These reward-combining methods are not invariant to such transformations, while the policy-aggregation methods we study are.

An entirely different notion of fairness in reinforcement learning is studied in Jabbari et al. (2017), where there is only a single reward function, but the notion of fairness is not for the final policy, but for the learning algorithm itself. That is, in an online-learning setting, each action taken during learning may involve real-life consequences (e.g. a hiring decision) and the goal is to make fair decisions in each state, while still rapidly converging to an optimal policy.

## 3 PRELIMINARIES

For a natural number $n$, we use the notation $[n] = \{1, \ldots, n\}$. A halfspace is a subset of $\mathbb{R}^n$ given by an affine inequality $\{x \in \mathbb{R}^n \mid w^\top x \le b\}$. A polytope $\mathcal{K} \subseteq \mathbb{R}^n$ is the intersection of finitely many halfspaces. The vertices of a polytope are the points which cannot be realized as the convex combination of two distinct points in the polytope. A simplex in $\mathbb{R}^n$ is the convex hull of $n + 1$ linearly independent points, which form the vertices of the simplex. The centroid $c_\mathcal{K}$ of a convex set $\mathcal{K}$ is the uniform average over the points in the set

$$c_\mathcal{K} = \frac{1}{\text{Vol}(\mathcal{K})} \int_\mathcal{K} x \, dx.$$

When $\mathcal{K} = \text{conv}(v_1, \ldots, v_{n+1})$ is a simplex, the centroid takes a particularly simple form, given by the average of the vertices $c_\mathcal{K} = \sum_{i=1}^{n+1} v_i$. An algorithm of Cohen & Hickey (1979) shows that any bounded polytope $\mathcal{K}$ can be decomposed into a union of simplices, such that any two simplices in the decomposition intersect in a face, and all vertices of the simplices are vertices of the original polytope. Furthermore, given any vertex $v_i$ of the polytope $\mathcal{K}$, the decomposition can be chosen such that $v_i$ is a vertex of *every* simplex in the polytope.

**Multi-objective Markov Decision Process.** The problem of producing a single policy to best represent the preferences of multiple agents is modelled as a Multi-Objective Markov Decision Process (MOMDP). An MOMDP $\mathcal{M}$ is given by a tuple $(\mathcal{S}, \mathcal{A}, \mathcal{P}, R_1, \ldots, R_n, \rho_0, \gamma)$, where $\mathcal{S}$ is a finite set of states, $\mathcal{A}$ is a finite set of actions, and $\mathcal{P} : \mathcal{S} \times \mathcal{A} \to \Delta(\mathcal{S})$ is the probability transition function, which gives the probability $\mathcal{P}(s' \mid s, a)$ of transitioning to state $s'$ when action $a$ is taken in state $s$. Each function $R_i : \mathcal{S} \times \mathcal{A} \to \mathbb{R}$ is the reward function corresponding to the $i$-th agent, $\rho_0 \in \Delta(\mathcal{S})$ is the initial state distribution, and $\gamma \in [0, 1]$ is the discount factor. A policy $\pi : \mathcal{S} \to \Delta(\mathcal{A})$ assigns a probability $\pi(a \mid s)$ of taking action $a \in \mathcal{A}$ in state $s \in \mathcal{S}$. A policy $\pi$ for an MOMDP $\mathcal{M}$ induces a probability distribution on sequences $s_0, a_0, s_1, a_1, \ldots$ where $s_0 \sim \rho_0$, $a_t \sim \pi(\cdot \mid s_t)$, and $s_{t+1} \sim \mathcal{P}(\cdot \mid s_t, a_t)$. We will use the notation $\text{Pr}_\pi$ and $\mathbb{E}_\pi$ to respectively denote the probability and expectation with respect to this random sequence. The *return* $J_i(\pi)$ for agent $i$ under policy $\pi$ is given by expected cumulative discounted reward with respect to $R_i$,

$$J_i(\pi) = (1 - \gamma) \mathbb{E}_\pi \left[ \sum_{t=1}^{\infty} \gamma^t R_i(s_t, a_t) \right]$$

where $s_0 \sim \rho_0$.

We also consider the setting of *average return*, where the return for agent $i$ under policy $\pi$ is given by the expected average reward

$$J_i^{\text{avg}}(\pi) = \lim_{T \to \infty} \frac{1}{T} \mathbb{E}_{\pi} \left[ \sum_{t=1}^{T} R_i(s_t, a_t) \right].$$

For $\epsilon > 0$, a policy $\pi$ in an MOMDP is $\epsilon$-*Pareto optimal* if for any other policy $\pi'$ there exists at least one agent $i$ such that $J_i(\pi') < J_i(\pi) + \epsilon$.

A *mixed policy* $\mu$ is a probability distribution over stochastic policies $\pi$. The return of a mixed policy $\mu$ is given by $J_i(\mu) = \mathbb{E}_{\pi \sim \mu}[J_i(\pi)]$.

**The state-action value occupancy measure.** While the return of agent $i$ under policy $\pi$ is a complicated function of $\pi$, it is a linear function of the *state-action value occupancy measure* $d_\pi : \mathcal{S} \times \mathcal{A} \to \mathbb{R}$ given by

$$d_\pi(s, a) = (1 - \gamma) \sum_{t=1}^{\infty} \Pr_{\pi} [s_t = s] \gamma^t \pi(a \mid s).$$

In particular $J_i(\pi) = \sum_{s,a} d_\pi(s, a) R_i(s, a) = \langle d_\pi, R_i \rangle$.

**Definition 3.1** (State-action value occupancy polytope). The *state-action value occupancy polytope* $\mathcal{O}$ for an MOMDP $\mathcal{M}$ is the set of all state-action value measures $d_\pi$ for some policy $\pi$ in the MOMDP $\mathcal{M}$. This set is a convex polytope given by the constraints

$$\mathcal{O} = \left\{ d_\pi \mid d_\pi \geq 0, \sum_a d_\pi(s, a) = (1 - \gamma)\rho_0(s, a) + \gamma \sum_{s', a'} \mathcal{P}(s \mid s', a')\pi(a' \mid s') \qquad \forall s \in \mathcal{S} \right\}.$$

## 4 FAIR REINFORCEMENT LEARNING OVER CONVEX SETS OF ALTERNATIVES

In classical social choice theory, there is a finite set $C$ of $m$ possible alternatives, and each agent $i$ has a strict ordering $\sigma_i : [m] \to C$ which means that for all $j < j'$ agent $i$ prefers $\sigma_i(j)$ to $\sigma_i(j')$. However, these methods cannot be directly applied to the fair reinforcement learning problem, as there is in fact a continuous set of stochastic policies $\pi$ in an MOMDP.

The policy aggregation framework of Alamdari et al. (2024) is based on viewing the state-action value occupancy polytope as the set of alternatives from which the $n$ agents will choose a policy. If $J_i(\pi) > J_i(\pi')$ then agent $i$ prefers policy $\pi$ to policy $\pi'$, and if $J_i(\pi) = J_i(\pi')$ then agent $i$ is indifferent between the two policies. Each return $J_i$ therefore induces a weak preference ordering on the policies $\pi$, and hence on the occupancy measures $d_\pi \in \mathcal{O}$. The overall goal is to compute a single policy in a way that is fair given the preference orderings of the $n$ agents induced by the rewards $R_i$. The individual preference orderings are invariant to positive affine transformations of the rewards $R_i$, and thus any algorithm which only takes as input the preference orderings will always output a policy invariant to the same set of transformations. This fact further allows us to assume that each individual reward function $R_i$ takes values in $[0, 1]$, as this can always be achieved by a positive affine transformation of the original reward.

In order to design fair aggregation policies over the occupancy polytope, we will first need a natural generalization of the notion of a ranking from the finite alternative case. In particular, note that an alternative $c \in C$ is ranked in the $j$-th position by agent $i$ if and only if a uniform random $c' \in C$ is weakly preferred to $c$ with probability $j/m$. This observation leads naturally to the following definition

**Definition 4.1** (Expected return distribution). Let $\mathcal{M}$ be an MOMDP and $\mathcal{D}$ be a probability distribution supported on the occupancy polytope $\mathcal{O}$. For $v \in \mathbb{R}$ the expected return distribution of agent $i$ is given by the cumulative distribution function $F_{i,\mathcal{D}}(v) = \Pr_{d_\pi \sim \mathcal{D}} [J_i(\pi) \leq v]$.

In words, the CDF $F_{i,\mathcal{D}}(v)$ encodes the total probability mass under $\mathcal{D}$ of policies that achieve returns at most $v$.

The notion of fairness that we consider in this paper can now be defined in terms of the quantiles of the expected return distributions for each agent.

**Definition 4.2.** Let $\mathcal{D}$ be a probability distribution over the occupancy polytope $\mathcal{O}$ of an MOMDP $\mathcal{M}$. For $q \in [0, 1]$ a policy $\pi$ is *q-quantile fair with respect to $\mathcal{D}$* in $\mathcal{M}$ if $J_i(\pi) \geq F_{i,\mathcal{D}}^{-1}(q)$ for all $i \in [n]$. A policy $\pi$ is *$\epsilon$-max-quantile* fair with respect to $\mathcal{D}$ if $\pi$ is $q$-quantile fair with respect to $\mathcal{D}$, and there does not exist any $(q + \epsilon)$-quantile fair policy.

Intuitively, $q$-quantile fairness means that the policy $\pi$ is preferred to at least a $q$-fraction of alternatives by every agent $i$ simultaneously. Definition 4.2 is a generalization of the definition of $q$-quantile fairness of Alamdari et al. (2024), which considers the special case where $\mathcal{D}$ is the uniform distribution over the occupancy polytope $\mathcal{O}$. Our overall goal is to design algorithms that achieve $q$-quantile fairness for as large a value of $q$ as possible, which we refer to as max-quantile fairness.

## 5 Fairness from distributions over the occupancy polytope

In this section we consider the question of how best to choose the distribution $\mathcal{D}$ over the occupancy polytope $\mathcal{O}$ that determines the notion of $q$-quantile fairness in Definition 4.2. We first show that choosing the uniform distribution over $\mathcal{O}$ can lead to computationally intractable estimation problems for estimating the quantile function $F_{i,\mathcal{D}}^{-1}$. We then show that choosing $\mathcal{D}$ to be uniform on the polytope of individually optimal occupancy measures can resolve this issue.

### 5.1 The drawbacks of uniform $\mathcal{D}$

In prior work $\mathcal{D}$ is chosen to be the uniform distribution over the entire state-occupancy polytope Alamdari et al. (2024). However, we will argue that this choice of $\mathcal{D}$ may not be particularly meaningful in the setting where non-trivial learning is necessary to find a policy that achieves non-trivial returns for each agent. For example, in typical deep reinforcement learning problems, a randomly selected policy will achieve some trivial level of rewards, but training can converge to a policy that does quite well and learns very complex behavior. Put another way, the number of samples required to randomly draw a policy that performs well in deep reinforcement learning is prohibitively large, and certainly much larger than the training budget of the typical RL algorithm.

In the scenario outlined above, less than a very small fraction of the occupancy polytope achieves non-trivial rewards, which implies that random sampling to estimate $F_{i,\mathcal{D}}$ will yield little to no information. We formalize this issue with the uniform distribution on $\mathcal{O}$ in the following theorem.

**Theorem 5.1.** *There exists an MOMDP $\mathcal{M}$ with $n \geq 2$ agents such that for every $\epsilon \geq \sqrt{2|S|/n^{|S|-1}}$ (1) There is a pareto optimal, $(1 - \epsilon)$-quantile fair policy $\pi'$ with respect to the uniform distribution on $\mathcal{O}$, such that one agent receives a $(1 - \epsilon)$ fraction of their maximum return, while all other agents receives less than a $\sqrt{|S|/(2n^{|S|+1})}$ fraction of their maximum return. (2) There is a pareto optimal, $(1 - \epsilon)$-quantile fair policy $\pi^*$ such that all agents receive a $\frac{1}{n}$ fraction of their maximum return.*

The proof of Theorem 5.1 appears in Section C. To summarize, there is an MOMDP where $q$-quantile fairness with respect to the uniform measure on $\mathcal{O}$ only provides meaningful constraints on the returns when $q$ is exponentially close to 1 as a function of the number of states. This is a significant problem for current algorithms, which require estimating $F_{i,\mathcal{D}}$ from random samples, and thus will have exponentially large sample complexity and runtime in order to achieve meaningful results using $q$-quantile fairness in the MOMDP of Theorem 5.1.

A further drawback of current methods is that, to even generate a single approximately uniform random sample from $\mathcal{O}$ requires running a Markov chain for sampling from convex bodies. However, running such an algorithm requires an explicit description of the polytope $\mathcal{O}$, which is not in general available for many MDPs of interest. In particular, such algorithms cannot be implemented efficiently when given access to only a policy optimization oracle for $\mathcal{O}$. Notably, the mixing time of such chains is polynomial in the dimension of the polytope, which for $\mathcal{O}$ is $O(|S| \cdot |A|)$. Such a mixing time is completely impractical in settings with very large state spaces, where policy optimization may nevertheless be possible.

## 5.2 Choosing $\mathcal{D}$ via individually optimal policies

The difficulties described above in choosing $\mathcal{D}$ to be the uniform distribution on $\mathcal{O}$ arise both because the set of occupancy measures achieving non-trivial rewards can have very small volume, and the task of even drawing a single uniform sample from $\mathcal{O}$ may be intractable. We propose avoiding these issues by choosing $\mathcal{D}$ to be the uniform distribution on a polytope derived from the occupancy measures of individually optimal policies. We will prove that this guarantees that the sample complexity necessary to estimate $F_{i,\mathcal{D}}$ for non-trivial fairness is bounded, and that there is an oracle-efficient sampling algorithm for $\mathcal{D}$. Our approach is to work in an embedding of a subset the occupancy polytope $\mathcal{O}$ defined by individually optimal policies. First, we define the reward embedding, which maps each occupancy measure $d_\pi$ to the $n$-dimensional vector given by the returns of each agent.

**Definition 5.2** (Reward embedding). Given an MOMDP $\mathcal{M}$, the *reward embedding* is the mapping $\Phi : \mathcal{O} \to \mathbb{R}^n$ given by $\Phi(d_\pi) = (J_1(\pi), \ldots, J_n(\pi))$.

The reward embedding captures the intuition that we only care about differences between two policies $\pi$ and $\pi'$ with regards to their returns for each agent $i$. Next, the main intuition is that we want to focus on $q$-quantile fairness over a subset of policies that achieves good rewards. A natural candidate is the polytope defined by the convex hull of individually optimal policies $\pi_i^*$ for each reward $R_i$ respectively.

**Definition 5.3** (Optimal reward embedding polytope). Let $\pi_i^*$ be an optimal policy for agent $i$ i.e. $\pi_i^* \in \arg\max_\pi J_i(\pi)$. The *optimal reward embedding polytope* induced by $\pi_1^*, \ldots \pi_n^*$ is the set $\mathcal{K}^* = \mathrm{conv}\left(\{\Phi(d_{\pi_i^*})\}_{i \in [n]}\right) \subseteq \mathbb{R}^n$ i.e. the reward embeddings of occupancy measures in the convex hull of the individually optimal occupancy measures.

Finally, we choose $\mathcal{D}$ to be the distribution on $\mathcal{O}$ induced by the uniform distribution on $\mathcal{K}^*$.

**Definition 5.4** (Optimal occupancy distribution). The optimal occupancy distribution is given by sampling a uniform random point $x \in \mathcal{K}$, and the sampling a uniform random occupancy measure $d_\pi$ from the set $\Phi^{-1}(x)$.

A few remarks are in order. First, an optimal reward embedding polytope $\mathcal{K}^*$ is invariant to positive affine transformations of the rewards, because the individually optimal policies $\pi_i^*$ are. Thus, letting $\mathcal{D}$ be the uniform distribution on $\mathcal{K}^*$ preserves this invariance, a key requirement for fair reinforcement learning. Second, in order to estimate the expected return distribution $F_{i,\mathcal{D}}$ it is not necessary to sample from $\Phi^{-1}(x)$. By construction, for each agent $i$ all the occupancy measures $d_\pi \in \Phi^{-1}(x)$ give the same return $x_i = \Phi(d_\pi)_i$. Thus, in order to estimate $F_{i,\mathcal{D}}$ one needs only to sample uniformly from $\mathcal{K}^*$.

In general, it may be that for some individually optimal occupancy measure, the embedding $\Phi(d_{\pi_i^*})$ is in the interior of the induced polytope $\mathcal{K}^*$, i.e. that $\Phi(d_{\pi_i^*})$ is a non-trivial convex combination of the embeddings of some other occupancy measures $\Phi(d_{\pi_j^*})$. However, the next proposition shows that every optimal reward embedding polytope can be assumed to be induced by individually optimal occupancy measures $d_{\pi_i^*}$ that correspond to vertices of $\mathcal{K}^*$.

**Proposition 5.5.** *Let $\pi_1^*, \ldots, \pi_n^*$ be a set of individually optimal policies, and $\mathcal{K}^*$ the induced optimal reward embedding polytope. Then there is a set $\pi_1', \ldots, \pi_n'$ of individually optimal policies such that every $\Phi(d_{\pi_i'})$ is a vertex of $\mathcal{K}^*$. Furthermore, this set can be efficiently computed by evaluating $J_i(\pi_j^*)$ for all $i, j$.*

The proof of Proposition 5.5 appears in Section A. We next show that choosing $\mathcal{D}$ to be uniform on $\mathcal{K}^*$ ensures the existence of a $q$-quantile fair policy $\pi$ achieving non-trivial returns for each agent.

**Theorem 5.6.** *Let $\pi_1^*, \ldots, \pi_n^*$ be individually optimal policies. For every MOMDP $\mathcal{M}$ there is a policy $\pi$ that is $\frac{1}{e}$-quantile fair with respect to the uniform distribution on $\mathcal{K}^*$. Furthermore, under $\pi$ every agent receives at least a $\frac{1}{n}$-fraction of their maximum possible return.*

The proof of Theorem 5.6 appears in Section B.

# 6 AN ORACLE-EFFICIENT ALGORITHM FOR MAX-QUANTILE FAIR REINFORCEMENT LEARNING

In this section we design an oracle-efficient algorithm for computing a max-quantile fair policy with respect to the optimal occupancy polytope, based on the multiplicative weights update method. In particular, our algorithm solves a series of standard RL policy optimization problems, where the single reward is given by a weighted combination $R = w_1 R_1 + \ldots w_n R_n$ at each step.

## 6.1 SAMPLING AND ESTIMATING QUANTILES

As a first step, we need an algorithm for estimating the quantile function $F_{i,\mathcal{D}}^{-1}$ when $\mathcal{D}$ is the optimal occupancy distribution. Such an algorithm follows directly from an efficient algorithm for sampling from the optimal occupancy distribution, which we introduce in this subsection. The main high-level point is that computing returns $J_j(\pi_i^*)$ for all $i$ and $j$ gives an explicit description of $\mathcal{K}^*$ as a polytope in $n$ dimensions, and so standard methods for sampling from a polytope apply. While in general this requires Markov chain Monte Carlo, we emphasize that in the case where the vectors $\Phi(d_{\pi_i^*})$ are linearly independent there is a fast, exact sampling method. This case is of particular interest, because for complex practical MDPs with noisy reward estimates, the vectors $\Phi(d_{\pi_i^*}) = (J_1(\pi_i^*), \ldots, J_n(\pi_i^*))$ will be linearly independent with high probability. See Proposition E.1 in Section E for a formal statement and proof. This is especially likely to be the case when the reward functions themselves are noisy estimates learned from preferences of the individual agents, as is standard practice in RLHF.

---

**Algorithm 1** Sampling from the optimal occupancy distribution

1: **Input:** MOMDP $\mathcal{M}$, desired number of samples $S$
2: Compute the $n$ individually optimal policies $\pi_i^* \in \operatorname{argmax}_\pi J_i(\pi)$ for $i \in [n]$.
3: If for any policy $\Phi(d_{\pi_i^*})$ is not a vertex of $\mathcal{K}^*$ use the method of Proposition 5.5 to replace it with an individually optimal policy that is a vertex.
4: **for** $s = 1$ **to** $S$ **do**
5:     **if** The vectors $\Phi(d_{\pi_i^*})$ are linearly independent. **then**
6:         Sample $n$ independent $\mathrm{Exp}(1)$ random variables $\alpha_j$ for $j \in [n]$.
7:         Let $\beta_j = \frac{\alpha_j}{\sum_{k=1}^n \alpha_k}$.
8:         Let $v_{i,s} = \sum_{j=1}^n \beta_j J_i(\pi_j^*)$ for $i \in [n]$.
9:     **else**
10:         Use the hit-and-run Markov chain to sample an approximately uniform point $x \in \mathcal{K}^*$.
11:         Let $v_{i,s} = x_i$
12:     **end if**
13: **end for**
14: **Return:** The $v_{i,s}$ which are $S$ samples for each of $n$ agents for the value $J_i(\pi)$ for $\pi \sim \mathcal{D}$.

---

Algorithm 1 gives the algorithm for sampling from $\mathcal{K}^*$. When the vectors $\Phi(d_{\pi_i^T})$ are linearly independent, $\mathcal{K}^*$ is a simplex with $n$ vertices. The uniform distribution on the simplex is well-known to be given by picking $n$ independent $\mathrm{Exp}(1)$ random variables, normalizing them to sum to one, and selecting the point given by the resulting convex combination of the vertices Devroye (2006). When $\mathcal{K}^*$ is a general polytope, the hit-and-run Markov Chain obtains an approximately uniform sample in $\widetilde{O}(n^3)$ iterations (Lovász, 1999), each of which requires solving $\widetilde{O}(1)$ $n \times n$ linear programs to determine membership of a point in $\mathcal{K}^*$ (see Section F). We give these results formally in Theorems 6.1 and 6.2 below.

**Theorem 6.1.** *If the vectors $\{\Phi(d_{\pi_i^*})\}_{i=1}^n$ are linearly independent then for each $i$ the set $\{v_{i,s}\}_{s=1}^S$ returned by Algorithm 1 is an exactly uniform random sample from $J_i(\pi)$ where $\pi \sim \mathcal{D}$. Further, the algorithm requires $n$ calls to a policy optimization oracle, and otherwise runs in $O(n^2)$ time.*

**Theorem 6.2.** *If the vectors $\{\Phi(d_{\pi_i^*})\}_{i=1}^n$ are not linearly independent, then for each $i$ the set $\{v_{i,s}\}_{s=1}^S$ returned by Algorithm 1 is an approximately random sample from $J_i(\pi)$ with $\pi \sim \mathcal{D}$. In this case the algorithm requires $n$ calls to a policy optimization oracle, and requires $\widetilde{O}(n^3)$ solves of an explicit linear program with $n$ variables and $n$ constraints.*

---

**Algorithm 2** Optimal occupancy quantile estimation

---
1: **Input:** MOMDP $\mathcal{M}$, $q \in [0, 1]$, target accuracy $\epsilon$
2: Draw $O\left((q(1 - q)/\epsilon)^2\right)$ samples $v_{i,s}$ using Algorithm 1.
3: For each $i$ sort $v_{i,s}$.
4: **Return:** $v_i^* = v_{i,\lfloor q \cdot s \rfloor}$ as an estimate of $F_{i,\mathcal{D}}^{-1}(q)$ for each $i \in [n]$.

---

**Algorithm 3** Oracle-efficient $q$-quantile fairness

---
1: **Input:** MOMDP $\mathcal{M}$, $q \in [0, 1]$, target welfare $U$, target accuracy $\epsilon$
2: Set $w_i^0 = \frac{1}{n}$ for $i \in [n]$, $u^0 = 0$.
3: Set $T = \log(n + 1)/\epsilon^2$, and $\eta = \sqrt{\log(n + 1)/T}$.
4: Use Algorithm 2 to get an estimate $v_i$ of $F_{i,\mathcal{D}}^{-1}(q)$ for $i \in [n]$.
5: **for** $t = 1$ **to** $T$ **do**
6:     Compute $\pi_t \in \text{argmax}_\pi \sum_i (w_i^t + u^t) J_i(\pi)$.
7:     Update $w_i^{t+1} = w_i^t \cdot \exp(\eta(v_i - J_i(\pi_t)))$.
8:     Update $u^{t+1} = u^t \cdot \exp\left(\eta\left(U - \sum_{i=1}^n J_i(\pi_t)\right)\right)$
9: **end for**
10: Let $\mu$ be the mixed policy given by the uniform distribution over $\pi_t$ for $t \in [T]$.
11: **Return:** If $\mu$ is $(q - \epsilon)$-quantile fair, and achieves total welfare $U - \epsilon$ return $\mu$, otherwise return "infeasible".

---

Finally, given the ability to sample from $\mathcal{D}$, we can use standard quantile estimators to compute $F_{i,\mathcal{D}}^{-1}$ from samples. Algorithm 2 gives pseudocode for quantile estimation. The most important point is the sample complexity of $O(q(1 - q)/\epsilon^2)$ required to estimate $F_{i,\mathcal{D}}^{-1}(q)$ to within accuracy $\epsilon$.

## 6.2 COMPUTING A MAX-QUANTILE FAIR POLICY

In this section we utilize the ability to efficiently estimate $F_{i,\mathcal{D}}^{-1}$ to design oracle-efficient algorithms achieving max quantile fairness. Algorithm 3 takes as input a values $q, U \in [0, 1]$ and returns a $(q - \epsilon)$-quantile fair mixed policy $\mu$ achieving total welfare at least $U - \epsilon$. The algorithm is an instance of the multiplicative weights update (MWU) method from online convex optimization. Algorithm 4 then uses Algorithm 3 as a subroutine to binary search for a welfare maximizing $\epsilon$-max-quantile fair mixed policy.

---

**Algorithm 4** Oracle-efficient max-quantile fairness

---
1: **Input:** MOMDP $\mathcal{M}$, target accuracy $\epsilon$
2: Binary search over $q \in [0, 1]$ for the maximum value $q^*$ such that running Algorithm 3 with $q = q^*$ and $U = 0$ does not return "infeasible".
3: Binary search over $U \in [0, 1]$ for the maximum value $U^*$ such that running Algorithm 3 with $q = q^*$ and $U = U^*$ does not return "infeasible".
4: Return the mixed policy $\mu$ output by Algorithm 3 when $U = U^*$ and $q = q^*$.

---

**Theorem 6.3.** *The mixed policy $\mu$ returned by Algorithm 3 is $(q - \epsilon)$-quantile fair, and achieves total welfare at least $U - \epsilon$.*

The proof appears in SectionD.

## 7 EXPERIMENTS

In this section we run experiments comparing our oracle-efficient max-quantile fairness algorithm to the original max-quantile fairness algorithm of Alamdari et al. (2024). Because we consider different distributions on policies for our definition of max-quantile fairness, the policies output by the two algorithms will in general be different. The environment is the same factory monitoring MDP used by Alamdari et al. (2024) (available under the MIT license), which is itself a modification of an environment introduced by D'Amour et al. (2020). The setup involves monitoring $m = 5$ warehouses

|  | Oracle-Efficient Max-Quantile | Original Max-Quantile | Egalitarian | Utilitarian |
|---|---|---|---|---|
| Nash Welfare | $0.701 \pm 0.015$ | $0.654 \pm 0.010$ | $0.4655 \pm 0.0581$ | $0.5736 \pm 0.0471$ |
| Gini Coefficient | $0.087 \pm 0.007$ | $0.066 \pm 0.003$ | $0.2126 \pm 0.0209$ | $0.2020 \pm 0.0182$ |
| $q^*$ | $0.883 \pm 0.055$ | $0.999 \pm 0.001$ | N/A | N/A |

Table 1: Nash welfare, Gini coefficient, for the oracle-efficient max-quantile fairness, original max-quantile fairness, egalitarian and utilitarian algorithms. Max-quantile $q^*$ is reported for the two max-quantile fairness algorithms.

to prevent incidents, where only one warehouse can be monitored in each time-step, and each of $n = 10$ agents incurs different costs for an incident in each warehouse. Warehouses can be in one of three states normal, risky, or incident. Every unmonitored warehouse has some probability of transferring from normal to risky, and from risky to incident. Monitoring a risky warehouse resets it to the normal state. At the beginning of each experiment, the transition probabilities and the agent costs are randomly sampled from the same set of values as used in Alamdari et al. (2024). The wallclock time to achieve max-quantile fairness via our method in this setting is approximately 10 minutes on a laptop with an Intel core ultra 7 and 32 GB of RAM. This contrasts with the reported runtime of approximately 2 hours utilizing 20 parallel threads on an AMD EPYC 7502 32-Core Processor with 258GiB system memory reported by the prior method of Alamdari et al. (2024).

Figure 1 shows the sorted returns for each of the agents under the two different algorithms. All agents, except the lowest return one, receive larger rewards under our oracle-efficient method. This is consistent with the intuition that choosing $\mathcal{D}$ to be induced by the uniform distribution on $\mathcal{K}^*$ will tend to lead the algorithm to consider higher-return policies for each agent.

In Table 1 we report the Nash welfare, Gini coefficient, and the maximum quantile $q^*$ achieved for the normalized agent returns for both our oracle-efficient algorithm and the original max-quantile fair algorithm. As a baseline point of comparison we also include the classic utilitarian algorithm (which maximizes the sum of the rewards), and the egalitarian algorithm (which maximizes the minimum reward). Both variants of max-quantile fairness outperform these baseline methods. Additionally, the results on $q^*$ confirm that the issue identified in Theorem 5.1 is present in this environment. Even with $5 \cdot 10^5$ randomly sampled policies, we get $q^* = 1 - \epsilon$ for $\epsilon = 0.001$, indicating that nearly all of the policies sampled uniformly from $\mathcal{O}$ achieve low returns. In contrast, the maximum quantile found by our oracle-efficient algorithm

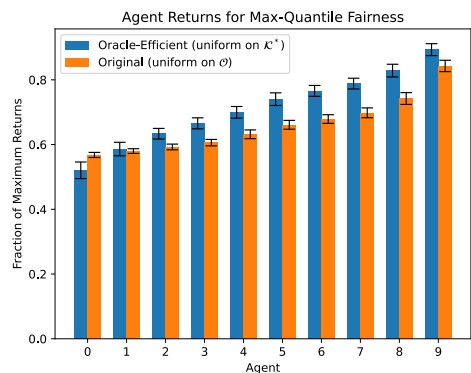

Figure 1: Sorted and normalized returns for each of 10 agents monitoring 5 factories.

is $q^* = 0.883$, indicating that under the uniform distribution on $\mathcal{K}^*$ one cannot do better than achieving returns in the top $12\%$ for each agent simultaneously. These results indicate that the sample-complexity for estimating $F_{i,\mathcal{D}}^{-1}$ via sampling from $\mathcal{K}^*$ is reasonable, while that of sampling from $\mathcal{O}$ grows rapidly.

## 8  DISCUSSION AND FUTURE WORK

We have designed an oracle-efficient algorithm for max-quantile fair reinforcement learning. That is, in any situation where standard single-reward reinforcement learning is possible, our algorithm provides a theoretically-founded method for producing a single policy that aggregates the preferences of multiple agents in a provably fair way. One main limitation of our work, is the question of how to obtain the individual reward functions in practical settings, especially for RLHF. Typical RLHF treats human preference rankings from multiple raters as noisy estimates of a single ground-truth reward, and it is an open question how best to learn estimates for multiple different rewards in a representative way. Our algorithm can be directly plugged into an RLHF pipeline where multiple reward models are available. One can simply use any existing policy optimization algorithm in place of the oracle from our algorithms.

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

## A   PROOF OF PROPOSITION 5.5

In this section we prove Proposition 5.5.

*Proof of Proposition 5.5.* Because $\mathcal{K}^* = \text{conv}\left\{\Phi(d_{\pi_i^*})\right\}_{i=1}^n$, every vertex of $\mathcal{K}^*$ is given by $\Phi(d_{\pi_i^*})$ for one of the individually optimal policies $\pi_i^*$. Let $j$ be the smallest index such that $\Phi(d_{\pi_j^*})$ is in the relative interior of $\mathcal{K}^*$, and let $\sum_{i \neq j} \alpha_i \Phi(d_{\phi_i^*}) = \Phi(d_{\phi_j^*})$ be a convex combination of the vertices of $\mathcal{K}^*$ that yields $\Phi(d_{\phi_j^*})$. Thus, by the definition of $\Phi$,

$$J_j(\pi_j^*) = \sum_{i \neq j} \alpha_i J_j(\pi_i^*).$$

By optimality of $\pi_j^*$ with respect to $J_j$, $J_j(\pi_i^*) \leq J_j(\pi_j^*)$ for all $i$. Therefore, for every $i$ with $\alpha_i \geq 0$ it must be that $J_j(\pi_i^*) = J_j(\pi_j^*)$. Choose any such $i$ with $\alpha_i \geq 0$, and let $\pi_j' = \pi_i^*$, and $\pi_i' = \pi_i^*$ for $i < j$. Now inductively applying the same argument yields the desired set of optimal policies $\pi_i'$ where each $\Phi(d_{\pi_i'})$ is a vertex of $\mathcal{K}^*$.

Testing whether a point is in the relative interior of a polytope, and computing the convex combination $\alpha_i$ witnessing this can both be solved via a linear program with variables $\alpha$ and $n \times n$ constraint matrix derived from the $n \times n$ matrix of returns $J_i(\pi_j^*)$ (see Section F). Hence, the overall algorithm runs in time polynomial in $n$ given access to an oracle for finding the optimal policies $\pi_j^*$.   □

## B   PROOF OF THEOREM 5.6

In this section we prove Theorem 5.6, which shows that $q$-quantile fairness over the optimal occupancy polytope enjoys improved fairness guarantees when compared to the entire state-action value occupancy polytope.

*Proof of Theorem 5.6.* The proof of Theorem 2 of Alamdari et al. (2024) shows via Grünbaums's inequality that any policy $\pi$ corresponding to the centroid of a polytope $\mathcal{K}$ achieves $\frac{1}{e}$-quantile fairness with respect to the uniform distribution on $\mathcal{K}$.

We now turn to the question of the return of each agent. By Proposition 5.5 the individually optimal policies $\pi_1^*, \ldots, \pi_n^*$ can be assumed to be vertices of $\mathcal{K}^*$. Cohen & Hickey (1979) proves that any polytope can be decomposed into a union of simplices with disjoint interiors such that the vertices of each simplex in the decomposition are vertices of the polytope. Furthermore, the decomposition can be chosen such that for any single vertex $v$ of the polytope, every simplex in the decomposition has $v$ as a vertex.

Thus, for any individually optimal policy $\pi_i^*$, the polytope $\mathcal{K}^*$ can be decomposed into a set of simplices $\Delta_k$, each of which has $\Phi(d_{\pi_i^*})$ as a vertex. Let $v_{i,k}$ denote the $i$-th vertex of the simplex $\Delta_k$ in the decomposition. The centroid $c_{\mathcal{K}^*}$ of the polytope $\mathcal{K}^*$ can therefore be written as

$$c_{\mathcal{K}^*} = \frac{1}{\text{Vol}(\mathcal{K}^*)} \int_{\mathcal{K}^*} x \, dx = \sum_k \frac{1}{\text{Vol}(\mathcal{K}^*)} \int_{\Delta_k} x \, dx$$

$$= \sum_k \frac{\text{Vol}(\Delta_k)}{\text{Vol}(\mathcal{K}^*)} \cdot \frac{1}{n} \sum_{i=1}^n v_{i,k}.$$

The final equality uses the fact that the centroid of a simplex is the average of its vertices. Since $\Phi(d_{\pi_i^*})$ appears as a vertex of each $\Delta_k$, we conclude the centroid of $\mathcal{K}^*$ can be written as a convex combination of the vertices, such that $\Phi(d_{\pi_i^*})$ receives weight at least $\frac{1}{n}$. Recall that by positive affine invariance we may assume that all returns are non-negative. Thus, the policy $\pi$ induced by any occupancy measure $d_\pi \in \Phi^{-1}(c_{\mathcal{K}^*})$ achieves return at least $\frac{1}{n} J(\pi_i^*)$.   □

## C   PROOF OF THEOREM 5.1

In this section we prove Theorem 5.1, demonstrating the sample-complexity issues with the original approach to estimating the expected return distribution.

*Proof of Theorem 5.1.* Define $\mathcal{M}$ as follows. There are $n$ agents identified with $[n]$. The state-space $\mathcal{S}$ is a chain $s_1, \ldots, s_m$ of $m$ states, and the action space has size $n$, denoted $\{a_1, \ldots, a_n\}$. For each state $s_j$ with $j \leq m - 1$, each action $a_i$ with $i \geq 2$ transitions to state $s_1$, and action $a_1$ transitions to state $s_{j+1}$, both yielding zero rewards for all agents. For state $s_m$ all actions transition to state $s_1$ and action $a_i$ yields reward 1 for agent $i$ and reward 0 for all other agents. The initial state distribution $\rho_0$ puts all probability mass on $s_1$.

A uniformly randomly chosen occupancy measure $d_\pi$ induces a policy that takes action $a_j$ in state $s_i$ with probability $p_{ij}$, where for each $i \in [m]$, the vector $(p_{i1}, \ldots, p_{in})$ is an independent, uniformly random sample from the standard simplex with $n$ vertices. Thus, $\mathbb{E}[p_{i1}] = 1/n$ for all $i$, because the centroid of the standard simplex on $n$ vertices is $(1/n, \ldots, 1/n)$. When starting from state $s_1$, the policy $\pi$ must select action $a_1$ $m$ times in a row to reach state $s_m$. Any other choice transitions the agent back to state $s_1$. Thus for any action $a_j$, we have $d_\pi(s_m, a_j) \leq \prod_{i=1}^{m} p_{i1}$. Hence $\mathbb{E}[d_\pi(s_m, a_j)] \leq \prod_{i=1}^{m} \mathbb{E}[p_{i1}] = n^{-m}$. Let $\epsilon \geq \sqrt{2mn/n^m}$. By Markov's inequality $\Pr[d_\pi(s_m, a_j) \geq \sqrt{1/(2mn \cdot n^m)}] \leq \epsilon$. But state $s_m$ is the only state where any agent can receive rewards, and the reward received is at most 1. Thus, $\Pr_{d_\pi \sim \mathcal{O}}[J_i(\pi) \geq \sqrt{1/(2mn \cdot n^m)}] \leq \epsilon$ for each agent $i$. Hence, the quantile function satisfies $F_{i,\mathcal{D}}^{-1}(1 - \epsilon) \leq \sqrt{1/(2mn \cdot n^m)}$.

Next, consider the policy $\pi'$ that takes action 1 in every state $s_i$ with $i \leq m$, and in state $s_m$ takes action 1 with probability $(1 - \epsilon)$ and a uniform random action in $\{a_2, \ldots, a_n\}$ otherwise. The policy $\pi'$ is pareto optimal, because it visits $s_m$ the only state with any rewards with maximum possible probability, and hence any deviation which increases one player's reward will decrease another's. Further, $\pi'$ is $(1 - \epsilon)$-quantile fair because every agent receives return at least $\epsilon/(m(n - 1)) > \sqrt{2/(mn \cdot n^m)} \geq F_{i,\mathcal{D}}^{-1}(1 - \epsilon)$.

Finally, the policy $\pi^*$ that takes action 1 in every state $s_i$ with $i \leq m - 1$ and a uniformly random action in state $s_m$ receives average return $\frac{1}{n|\mathcal{S}|}$ for all agents $i$, which is a $\frac{1}{n}$-fraction of the maximum possible average return of $\frac{1}{|\mathcal{S}|}$ for each agent. $\qquad\square$

## D    PROOF OF THEOREM 6.3

In this section we provide the proof of Theorem 6.3 regarding the correctness of Algorithm 4.

*Proof of Theorem 6.3.* We first show that our algorithm is an instance of the multiplicative weights update (MWU) method, and the proof then follows from standard regret bounds for MWU (Freund & Schapire, 1997; Hazan et al., 2016). The $n + 1$ values $w_i^t, u^t$ are the weights. The losses in each step are given by $l_i^t(\pi_t) = J_i(\pi_t) - v_i$ corresponding to the $w_i^t$ weights, and $c^t(\pi_t) = \sum_{i=1}^{n} J_i(\pi_t) - U$ corresponding to the $u^t$ weight. Let $W^t = u^t + \sum_{i=1}^{n} w_i^t$ be the total weight in step $t$. For any policy $\pi$, the expected loss in step $t$ given by the weights is then

$$L(w^t, u^t, \pi) = \frac{1}{W^t} \left( u^t c^t(\pi) + \sum_{i=1}^{n} w_i^t l_i^t(\pi) \right)$$

$$= \frac{1}{W^t} \sum_i (w_i^t + u^t) J_i(\pi) - \frac{1}{W^t} \left( u^t U + w_i^t v_i \right)$$

The second term above has no dependence on $\pi$, and the scaling factor of $\frac{1}{W^t}$ also does not affect the maximum over $\pi$, hence

$$\underset{\pi}{\arg\max} \, L(w^t, u^t, \pi) = \underset{\pi}{\arg\max} \sum_i (w_i^t + u^t) J_i(\pi). \qquad (1)$$

Let us assume that there exists a policy $\pi^*$ that is $q$-quantile fair, and achieves total welfare $U$. Otherwise, Algorithm 3 will necessarily output "infeasible". The regret bounds for MWU imply that for any set of weights $w^*, u^*$,

$$\frac{1}{T} \sum_{t=1}^{T} L(w^t, u^t, \pi^t) - \frac{1}{T} \sum_{t=1}^{T} L(w^*, u^*, \pi^t) \leq \frac{\log(n + 1)}{\sqrt{T}} \qquad (2)$$

In each step of Algorithm 3, (1) implies that $\pi_t \in \text{argmax}_\pi L(w^t, u^t, \pi)$. Thus,

$$\frac{1}{T}\sum_{t=1}^{T} L(w^t, u^t, \pi^t) \geq \frac{1}{T}\sum_{t=1}^{T} L(w^t, u^t, \pi^*) \geq 0$$

where the final inequality follows from the fact that $\pi^*$ is both $q$-quantile fair, and achieves total welfare $U$, i.e. $l_i^t(\pi^*) \geq 0$ and $c^t(\pi^*) \geq 0$. Hence, the regret bound (2) implies that

$$\frac{1}{T}\sum_{t=1}^{T} L(w^*, u^*, \pi^t) \geq -\frac{\log(n+1)}{\sqrt{T}} = -\epsilon$$

for any choice of weights $w^*, u^*$. For each $i$, specializing the above inequality to the $n$ cases where $w_i^* = 1$, $w_j^* = 0$ for $j \neq i$, and $u = 0$, implies that the mixed policy $\mu$ satisfies $J_i(\mu) \geq v_i - \epsilon$. Specializing to the case $w_i = 0$ for all $i$, and $u = 1$, implies that $\mu$ satisfies $\sum_i J_i(\mu) \geq U - \epsilon$ as desired. $\square$

## E  LINEAR-INDEPENDENCE FOR NOISY ESTIMATES OF $\{\Phi(d_{\pi_i^*})\}_{i=1}^n$

In this section we provide a formal statement and proof showing that rewards estimated by sampling multiple runs of a policy in a stochastic MDP will lead to linear independence of the $\{\Phi(d_{\pi_i^*})\}_{i=1}^n$ with high probability. This is of significant practical relevance, because computing the expected rewards of standard deep reinforcement learning policies is essentially always done by computing the empirical mean of multiple sampled evaluation runs, which have non-trivial variance.

**Proposition E.1.** *Assume that each entry of the vectors $\{\Phi(d_{\pi_i^*})\}_{i=1}^n$ is estimated by taking multiple independent samples of the stochastic reward $J_j(\pi_i^*)$, and these stochastic reward estimates are continuous random variables with non-zero variance $\sigma_{i,j}$. Then this estimate of $\{\Phi(d_{\pi_i^*})\}_{i=1}^n$ converges to a distribution on vectors that are linearly independent with probability 1.*

*Proof.* Note that each coordinate of each $\Phi$ vector is an independent stochastic estimate of the returns of policy $\pi_i^*$ under reward function $J_j$ computed by averaging the rewards over multiple independent runs. Each such estimate converges to a Gaussian distribution with variance $\sigma_{i,j} > 0$ by the central limit theorem. Thus, the matrix obtained by stacking all the $\Phi$ vectors together is equal to an $n \times n$ matrix of the true mean returns, plus independent Gaussian distributed noise with non-zero variance added to each coordinate. Such a matrix is full rank with probability 1 (see for example Rudelson & Vershynin (2008)), and hence the $\Phi$ vector estimates converge to a distribution that is linearly independent with probability 1. $\square$

## F  ADDITIONAL DETAILS ON LINEAR PROGRAMS FOR MEMBERSHIP IN $\mathcal{K}^*$

Both Proposition 5.5 and Algorithm 1 rely on solving an $n \times n$ linear program for the problems related to membership of a point in $\mathcal{K}^*$. In this section we provide the linear programs in question. First, for Proposition 5.5, it is necessary to determine if a point $\Phi(d_{\pi_i^*})$ is a convex combination of the other vertices of $\mathcal{K}^*$, and if so to determine the weights $\alpha_j$ of the convex combination. This problem can be expressed via the following system of linear inequalities in the variables $\alpha_j$ for $j \in [n]$:

$$\sum_{j \neq i} \alpha_j \Phi(d_{\pi_j^*}) = \Phi(d_{\pi_i^*})$$
$$\sum_{j \neq i} \alpha_j = 1$$
$$\alpha_j \geq 0 \qquad \forall j \in [n]$$

which is the problem of finding a feasible solution to an $n \times n$ linear program.

Similarly, the hit-and-run walk used for sampling in Algorithm 1 in the case that $\mathcal{K}^*$ is not a simplex relies on the ability to test whether a given point $v \in \mathcal{K}^*$. This problem can be solved by the following

$n \times n$ linear program:

$$\sum_{j=1}^{n} \alpha_j \Phi(d_{\pi_j^*}) = v$$

$$\sum_{j=1}^{n} \alpha_j = 1$$

$$\alpha_j \geq 0 \qquad \forall j \in [n].$$

