# OpenReview forum: "Fair Reinforcement Learning for Just AI"
_ICLR.cc/2026/Conference — ICLR 2026 Poster_

### Official Review · Reviewer_whs3 · 2025-10-28

**Soundness:** 3
**Presentation:** 3
**Contribution:** 3
**Rating:** 8
**Confidence:** 4

**Summary:**

The paper studies quantile fairness in the context of reinforcement learning. The authors show intrinsic scalability limitations of previous work and propose a slightly different setting where the quantile fairness is measured wrt to the set of optimal policies for different agents rather than all possible policies. This alternative formulation allows for efficient algorithms to identify max-quantile fair policies. The proposed algorithm is evaluated in a synthetic factory monitoring environment and its effectiveness is assess across several different metrics (e.g., Nash welfare, Gini coefficient).

**Strengths:**

* The "negative" result showing that quantile fairness defined on all policies can be vacuous is very interesting. The intuition behind this result is intuitive (i.e., in a high-dimensional policy space, it is likely that many policies have poor reward, thus making the max-quantile fairness to be very close to 1.0) and sound and it is likely to hold in general. It is then supported by a theoretical derivation, which relies on the construction of a specific worst-case MDP. This result provides a very solid grounding to the need of developing alternative, more practical formulations.
* The formulation proposed in the paper (i.e., focusing on optimal policies) is sound and it makes the overall setting much more tractable. This is supported both theoretically (Thm 5.6 / 6.1 / 6.2) and empirically (L448) and it builds on a neat result showing how sampling from the policy space which is the image of the convex hull of optimal rewards can be done efficiently.
* The empirical results provide a good evidence of the properties of the proposed formulation and algorithm. While a direct comparison with the original quantile fairness formulation cannot be done (i.e., they optimize different objectives), the authors study a number of correlated metrics that support the theoretical claims: 1) the method is more efficient; 2) the max-quantile fairness is significantly lower, showing that the method makes a more "significant" trade-off between preferences of different agents; 3) Nash and Gini are better; 4) per-agent return is mostly better.

**Weaknesses:**

* The paper is originally motivated by RLHF and the use of deepRL techniques. While the current paper is clearly a significant step forward in that direction (e.g., the use of an oracle policy optimization routine, instead of assuming perfect knowledge of the underlying MDP and a more efficient algorithm), it still remains "limited" to a more theoretical setting without any direct evidence that the proposed method can scale to deepRL and can be applied to RLHF.

**Questions:**

* In the formulation of Sect.3 you assume a finite MDP, but later you only assume access to a policy optimization "oracle". Could you please clarify where the finite MDP assumption is needed in the algorithm and theoretical results?
* In L448 you make a high-level comparison on the running time of your algorithm and the previous one. Would it be possible to have a more direct comparison on the same hardware?
* When I first read the high-level description of the proposed formulation I was concerned that restricting the focus on agent-optimal policies would impose a too serious restriction. For instance, we can imagine problems where a fair policy should trade-off between the rewards of different agents resulting in a behavior that is fairly different than any of the optimal policies. While the definition of optimal occupancy distribution (Def 5.4) could be still quite rich, it is not easy to interpret. I have some questions in this respect
** Would it be possible to create "worst-case" MDPs where the max-quantile fair policy for the optimal occupancy distribution has q*~=0, i.e., the opposite case than in Thm5.1, where basically you restrict the policy space so much that no fair policy is possible.
** In the worst case of Thm5.1, can you prove that your formulation would have q*>>0?
** Can you provide some qualitative examples where your formulation is "meaningful", i.e., it doesn't exclude policies that are intuitively fair? The empirical results provide some evidence of that, but having a more intuitive illustration would help.

Minor
* L175: it should be J_i(pi')<J_i(pi) - eps?
* L300: \mathcal K -> \mathcal K^*
* Alg1: I would suggest to change the letter in "s=1 to S" to avoid confusion with states

---

> ### Author Response · Authors · 2025-11-23
>
> We are truly glad that you found our paper’s results neat, interesting, intuitive and sound while providing  a very solid grounding to the need of developing alternative, more practical formulations with further empirical results that provide good evidence of the properties of our proposed formulation and algorithm. We sincerely appreciate your in-depth and thorough review.
>
> 1. *“In the formulation of Sect.3 you assume a finite MDP, but later you only assume access to a policy optimization "oracle". Could you please clarify where the finite MDP assumption is needed in the algorithm and theoretical results?”*
>
> The assumption of finiteness is only needed to avoid handling technicalities arising from measure-theoretic considerations in continuous spaces. We believe all the results could be made to work with probability measures on $\mathbb{R}^d$, at the cost of additional notational and technical complexity. For example, the rewards would need to be measurable functions, the state-action value occupancy set $\mathcal{O}$ would be a convex subset in an infinite-dimensional space corresponding to some set of measures on $\mathbb{R}^d$ etc. In general, we believe the key algorithmic insights are well-captured by the case of a finite MDP. Further, when actually performing practical learning in MDPs, finite computer precision in representations of states and actions mean that we are in the finite MDP case.
>
> 2. *“In L448 you make a high-level comparison on the running time of your algorithm and the previous one. Would it be possible to have a more direct comparison on the same hardware?”*
>
> Running the method of Almadari et al on our hardware (Intel core ultra 7, 32 GB of RAM) took around 5 and half hours. In contrast, our method took 10 minutes. The original work of Almadari et al reports a running time of approximately 2 hours for their method on more powerful hardware. The main bottleneck of our hardware relative to the original results reported in Almadari et al is in the level of parallelism possible. Our hardware has 8 CPU cores, while theirs had 32.
>
> 3. *“When I first read the high-level description of the proposed formulation I was concerned that restricting the focus on agent-optimal policies would impose a too serious restriction. For instance, we can imagine problems where a fair policy should trade-off between the rewards of different agents resulting in a behavior that is fairly different than any of the optimal policies. While the definition of optimal occupancy distribution (Def 5.4) could be still quite rich, it is not easy to interpret. I have some questions in this respect ** Would it be possible to create "worst-case" MDPs where the max-quantile fair policy for the optimal occupancy distribution has q~=0, i.e., the opposite case than in Thm5.1, where basically you restrict the policy space so much that no fair policy is possible. ** In the worst case of Thm5.1, can you prove that your formulation would have q>>0? ** Can you provide some qualitative examples where your formulation is "meaningful", i.e., it doesn't exclude policies that are intuitively fair? The empirical results provide some evidence of that, but having a more intuitive illustration would help.”*
>
> We believe our Theorem 5.6 answers your main questions regarding our formulation. Theorem 5.6 proves that for our formulation we can always find a $q$-quantile fair policy with $q \geq 1/e \approx 0.368$ in **any MDP**. This follows from a classical result known as Grunbaums’ inequality, which shows that a halfspace through the centroid of a convex set divides it into two pieces, each of which have total measure at least $1/e$. The second part of Theorem 5.6 proves that in our formulation every agent can always get at least a $1/n$ fraction of their maximum total reward. In contrast, Theorem 5.1 showed that in the prior formulation, achieving $(1-\epsilon)$-quantile fairness for exponentially small $\epsilon$ is still not enough to guarantee each player more than an exponentially small fraction of their maximum total reward.
>
> We highly appreciate that our efforts to provide rigorous analysis and contributions were recognized by you and thank you very much for providing an insightful and thorough review.

---

### Official Review · Reviewer_Mguy · 2025-10-30

**Soundness:** 3
**Presentation:** 3
**Contribution:** 2
**Rating:** 4
**Confidence:** 4

**Summary:**

The paper studies a fair multi-objective Markov decision process problem that aims to compute a policy that is fair with respect to the objectives. The paper first introduces a q-quantile fairness notion based on the convex hull of optimal policies for individual objectives, and the authors argue that this notion is more appropriate than the one introduced in previous work. The paper then shows that this new notion makes it possible to find fair policies efficiently using a black-box policy optimization algorithm as its subroutine. Experiments were conducted to evaluate the fairness guarantee the algorithm yields.

**Strengths:**

The paper is clearly and nicely written. The results appear solid, and the exposition maintains a good balance between accessibility and mathematical rigor.

**Weaknesses:**

1. I don't see a strong connection between the model in the paper and RLHF. Most RLHF models are not built on MDPs.

2. Some of the problem setups feel somewhat artificial, potentially constructed for the sake of deriving results. For example, instead of looking at the quantile, why not simply look for solutions that yield x fraction of the maximum attainable value for every agent, and then maximize x? This seems much more straightforward and would simplify much of the analysis, and I don't see any obvious reason why this measure would be less desirable than the one proposed in the paper. In fact, the statement in Theorem 5.1 seems to suggest that we should care more about the fraction of each agent's maximum return.

3. In the paragraph at the top of Page 6, it is stated that the main drawback of existing methods is that they cannot handle large state spaces, whereas policy optimization is still feasible for large state spaces. I am not sure this is a valid argument. While it is true that there are policy optimization algorithms applicable to large state spaces, they generally do not yield exact optimal solutions. This seems critical as the main algorithm in this paper relies on a policy optimization algorithm that computes exact optimal solutions as its subroutine.

4. The sampling methods in Section 6.1 largely follow trivially from existing results.

5. Though the paper is overall clear, I find that it would still benefit from additional explanations and intuition in certain parts. Such as the 1/e-quantile fair guarantee in Proposition 5.5, and MWU-based algorithm (Algorithm 3).

**Questions:**

- Can you respond to the first three weaknesses I mentions?

- In Defintion 3.1, should $\rho_0(s,a)$ be $\rho(s)$?

- Line 6 of Algorithm 3, should it be $(\sum_i w_i^t + u^t) J_i(\pi)$?

---

> ### Author Response · Authors · 2025-11-23
>
> We are glad that you found our paper’s results solid while the exposition maintains a good balance between accessibility and mathematical rigor. Thank you for very much finding our paper clearly and nicely written.
>
> 1. *“I don’t see a strong connection between the model in the paper and RLHF. Most RLHF models are not built on MDPs.”*
>
> This is not quite correct. RLHF is a *special case* of reinforcement learning in general MDPs. In the case of RLHF the states are sequences of tokens $s$, the actions are individual tokens $a$, and given state $s$ and action $a$, then next state $s’$ is given by appending the token $a$ to the sequence $s$. Algorithms for RLHF are precisely standard reinforcement learning algorithms applied to this MDP. Thus, our methods directly apply to RLHF where there are multiple reward functions corresponding to a heterogeneous population of human preference rankings.
>
> 2. *“Some of the problem setups feel somewhat artificial, potentially constructed for the sake of deriving results. For example, instead of looking at the quantile, why not simply look for solutions that yield x fraction of the maximum attainable value for every agent, and then maximize x? This seems much more straightforward and would simplify much of the analysis, and I don't see any obvious reason why this measure would be less desirable than the one proposed in the paper. In fact, the statement in Theorem 5.1 seems to suggest that we should care more about the fraction of each agent's maximum return.”*
>
> First, the fairness notion you suggest (maximizing the minimum fraction of reward received by any agent) is known as egalitarian fairness, which we test as a baseline in our experiments. The results in Table 1 demonstrate that egalitarian fairness significantly underperforms $q$-quantile fairness.
> Second, the setting we study of $q$-quantile fairness, satisfies the key property of invariance to positive affine transformations of the rewards, whereas the egalitarian fairness notion you suggest does not. This key mathematical difference is what leads to the advantages for $q$-quantile fairness that we observe in the experiments.
>
> In more detail, invariance to positive affine transformations means that if we replace one reward function $r$ with $r’ = a\cdot r + b$ for $a>0$, then the fairness metric does not change. This property is of critical importance in the setting where rewards are learned from preference rankings (as in RLHF) or from agent behavior. This is because learning from preference rankings or agent behavior can only recover the rewards up to a positive affine transformation, meaning that preference rankings/behavior under $r$ and $a\cdot r + b$ will be exactly the same. Therefore it does not make sense to normalize and quantitatively compare two different reward functions learned from preferences or behavior, as the relative scales of the true reward functions of each agent are unknown. All that is known for sure is the relative ranking that each reward function assigns to a given policy $\pi$, and this is the only information that is used to determine $q$-quantile fairness. This key difference is also discussed on lines 122 -127 in the paper.
>
> 3. *“In the paragraph at the top of Page 6, it is stated that the main drawback of existing methods is that they cannot handle large state spaces, whereas policy optimization is still feasible for large state spaces. I am not sure this is a valid argument. While it is true that there are policy optimization algorithms applicable to large state spaces, they generally do not yield exact optimal solutions. This seems critical as the main algorithm in this paper relies on a policy optimization algorithm that computes exact optimal solutions as its subroutine.”*
>
> The main algorithm in the paper actually applies directly to approximate policy optimization as well. In particular, suppose one can guarantee $\alpha$-approximate policy optimization (i.e. the ability to compute a policy that achieves cumulative reward $J^* - \alpha$). Then the proof of Theorem 6.3 instead guarantees that Algorithm 3 outputs a policy that is $q- \epsilon - \alpha$-quantile fair, with total welfare $U - \epsilon -\alpha$. This is one of the strengths of the multiplicative weights update framework, which is inherently robust to approximation errors.
>
> Similarly, if Algorithm 1 is initialized with approximately optimal policies $\pi\_i$, the proof of Theorem 5.6 will instead guarantee that each agent receives at least a $1/n$ fraction of their approximately maximum rewards given by $\pi\_i$. This is the natural bound that one would hope for, given that one only has access to approximate policy optimization in this case. We can add a discussion of these results on approximation in the camera-ready version of the paper.

---

> ### Author Response · Authors · 2025-11-23
>
> 4. *”Line 6 of Algorithm 3, should it be $(\sum\_i w\_i^t + u^t)(J\_i(\pi))$”*
>
> No, line 6 of Algorithm 3 is correct as written. It is a weighted sum over the $n$ different returns $J\_i(\pi)$, each of which corresponds to the $i$-th reward function.

---

### Official Review · Reviewer_maiB · 2025-10-31

**Soundness:** 3
**Presentation:** 3
**Contribution:** 3
**Rating:** 8
**Confidence:** 3

**Summary:**

This paper studies fairness in reinforcement learning with large state-action spaces. More specifically, the work studies a setting with access to a policy optimization oracle and the goal is to obtain q-quantile fairness. The paper provides an algorithm based one multiplicative weights as well as theoretical fairness guarantees. Finally, the algorithm is experimentall evaluated on a tabular MDP.

**Strengths:**

**Clarity**
* The paper is well written and easy to follow.

**Motivation**
* The paper tackles an interesting, understudied problem that is of relevance in the time of finetuning LLMs using RL.

**Related Work**
* The treatment of related work is decent.

**Novelty**
* I am largely not aware of other work tackling similar problems and I believe that the work is sufficiently novel. One paper that should be highlighted is [1] which seems quite related. They also use an oracle notion in a multi-objective setting to obtain fairness guarantees. The fairness guarantees are slightly different but the manuscript might benefit from differentiating itself from this work (see Q1).

**Theoretical results**
* I mostly followed the arguments in the paper but did not verify the details of the proofs and parameter settings in the Appendix. However, using online learning techniques in combination with binary search to solve constraint optimization problems in large spaces is quite common and the algorithms as well as corresponding results seem sound.

**Experimental results**
* The experiments provide some nice intuition into the functionality of the algorithm.

[1] Intersectional Fairness in Reinforcement Learning with Large State and Constraint Spaces. Eaton et al. ICML 2025.

**Weaknesses:**

**Clarity**
* One confusing point to me was the terminology “policy aggregation”. In the introduction, the text refers to rewards that belong to policies but a reward is always defined with respect to an MDP and it refers to optimal policies for each agent. In hindsight, it is obvious what this is referring to but not on a first read of the intro. One needs to read section 3 to understand this part. I think it might make sense to clarify the relationship between agents and rewards in the intro when stating the model.

**Empirical Design and Evidence**
* The fundamental motivation of the paper is to design algorithms that work in high dimensional settings with neural networks but the experiments are conducted solely on what seems to be a tabular MDP. The manuscript could be strengthened by experiments actually using the complex function approximation objects that the algorithm is designed for.

Overall, the pros outweigh the cons in my eyes and I even though I would like to see some deep learning experiments, prior work seems to have accepted without them (e.g. [1]). I can't find much else to complain about and think this is a solid paper. Thus, I'm willing to err on the side of optimism an recommend acceptance.

**Questions:**

Q1: Can you elaborate on how your work differs from [1].

Q2: Maybe I'm missing something but what happens when 2 agent reward functions oppose each other and there exists no fair allocation?

---

> ### Author Response · Authors · 2025-11-23
>
> We are truly glad to hear that you found that our paper provides novel contributions and tackles an interesting, understudied problem that is of relevance in the time of fine tuning LLMs using RL. We highly appreciate that our efforts to provide solid contributions in this front were recognized and thank you very much for finding our paper well written and easy to follow with further experiments providing some nice intuition into the functionality of the algorithm. We highly appreciate your in-depth and insightful review.
>
> 1. *“Can you elaborate on how your work differs from [1]”*
>
> There are two major differences between the work of [1] and our work: first, how individual preferences are modeled in the MDP, and second, the notion of fairness used to aggregate preferences.
>
> In more detail: first, [1] considers a specific subclass of fair RL problems, where each state $s$ corresponds to an individual (or group of individuals), and the rewards obtained in state $s$ accrue to the corresponding individual. The goal in this case is to ensure fairness by maximizing the minimum reward of a group of individuals represented by a subset of states. Here the subsets in question are all intersections of some prespecified groups e.g. demographic subsets of a population and their intersections. In contrast, in our model there are a set of different reward functions, each of which can be thought of as corresponding to an individual or group, each having preferences regarding the entire rollout of the policy across states. Thus, our setting models a complex policy in an arbitrary state space (e.g. an economic policy for a country, a policy for the investments of publicly traded company, a policy for an LLM that interacts with millions of individuals), whereas the model of [1] focuses on a state space corresponding to individuals, where the policy takes an action for each individual (e.g. distributing disaster relief across an existing road network, distributing hospital beds or ventilator access across different patients).
>
> Second, [1] considers a notion of fairness where the goal is to maximize the minimum reward received by a protected group. As discussed in the related work section lines 122-127, maximizing the minimum reward inherently assumes that utilities/rewards can be numerically compared across individuals or groups. However, in the setting where individual utilities/rewards are learned (e.g. RLHF), each one can only be learned up to a positive affine transformation of the form $a\cdot r + b$ where $a>0$. That is, it is not possible to tell from an individual's behavior or preference rankings whether $r$ or $100\cdot r + 2$ is their true reward function. Thus, in this practically relevant setting, rewards cannot be numerically compared, and one must resort to methods that are invariant to positive affine transformations. Our model of policy aggregation is invariant to positive affine transformations, and hence differs significantly from the fairness model of [1], leading to quite different algorithmic challenges. In fact, in our experimental results in Table 1, the egalitarian baseline method exactly corresponds to maximizing the minimum rewards of an individual agent. As can be seen from these results, our notion of max-quantile fairness achieves a much fairer outcome as measured both by Gini coefficient and Nash welfare when compared to the egalitarian method of maximizing the minimum rewards.
>
> 2. *“Maybe I’m missing something but what happens when two agent reward function oppose each other and there exists no fair allocation?”*
>
> The power of $q$-quantile fairness is that, as shown in Theorem 5.6 no matter what the reward functions are, there is always a $1/e$-quantile fair policy where each agent receives at least $1/n$ fraction of their maximum return. For a simple example, suppose there are two reward functions, one which is 1 for action $a$ and 0 for action $b$, while the other is 1 for $b$ and 0 for $a$. Clearly the reward functions directly oppose each other, but the policy of choosing $a$ with 50% probability and $b$ with 50% probability is a natural way to be “fair” even in this situation. Thus, fairness is not about giving everyone their maximum, but about balancing all the competing rewards of the different players simultaneously. The notion of $q$-quantile fairness provides a formal way of guaranteeing this sort of fairness in arbitrary MDPs, not just in the simple example provided above.

---

### Official Review · Reviewer_tKms · 2025-11-10

**Soundness:** 2
**Presentation:** 3
**Contribution:** 3
**Rating:** 6
**Confidence:** 3

**Summary:**

This paper studies the problem of fair policy aggregation, using a quantile notion of fairness. The goal is to find a policy that each agent prefers to a k fraction of alternatives, for the largest feasible k. They build on prior work that addresses this problem for a broader notion of alternatives (the uniform distribution over all policies), but under a much stronger assumption of access to the MDP (known transition dynamics). They give an oracle-efficient algorithm for learning a quantile fair policy with respect to the occupancy measure polytope induced by the optimal policy for each agent, without any assumption of known dynamics or generative model.

They further show that redefining the set of alternative policies from prior work is necessary in their setting, as efficient algorithms to even estimate quantiles without knowledge of transition dynamics cannot exist if the uniform distribution over all policies is taken as the reference class.

**Strengths:**

This paper gives efficient, provable algorithms for a reasonable fairness notion for preference aggregation. They cleverly set up the comparison class of policies such that their sampling algorithm for estimating quantiles is efficient and their main algorithm for aggregation is a simple application of multiplicative weights.

**Weaknesses:**

The paper claims only $O(n)$ calls to a policy optimizer are required, but it appears that Algorithm 3 makes $T = \log(n+1)/\varepsilon^2$ calls to an optimization oracle, in addition to the $O(n)$ calls required to define the polytope.

“The key insight of Alamdari et al. (2024) is that one can still compute a continuous analogue of a “ranking” for each agent, by assigning to each policy a score of q if the agent receives a higher score than an q-fraction of all possible alternative policies” -- I think the second "score" here should be "reward".

**Questions:**

Please see weaknesses.

---

> ### Author Response · Authors · 2025-11-23
>
> We are truly glad to hear that you found that our paper provides clever, efficient, provable algorithms for a reasonable fairness notion for preference aggregation.
>
> 1. *“The paper claims only $O(n)$ calls to a policy optimizer are required, but it appears that Algorithm 3 makes $T = \log (n + 1)/\epsilon^2$ calls to an optimization oracle, in addition to the $O(n)$ calls required to define the polytope.”*
>
> In the case where $\epsilon$ is a fixed constant independent of $n$, then $O(n) + \log (n+1)/\epsilon^2 = O(n)$. This parameter regime is relevant where $n$ is very large e.g. voters in an election, individuals participating in an online marketplace etc. Of course, as you point out, $O(n) + \log (n+1)/\epsilon^2$ gives a more detailed trade-off of all the relevant problem parameters, so we will happily clarify this in the final draft.
>
> 2. *“I think the second ‘score’ here should be ‘reward’.”*
>
> Thank you for pointing out this typo, we will fix it.

---

### Meta-Review · Area_Chair_qTEH · 2025-12-22

**Summary:**

**Paper Summary**: This paper considers a fair policy aggregation problem. The paper considers a quantile fairness notion where the goal is to find a policy that will be at least $q$-quantile of the cumulative rewards across different objectives over some occupancy distribution. Their overall goal is to design an approach that will be invariant under an affine reward shift, as it is important when we collect data under preference ordering. The paper first showed that if the occupancy distribution is uniformly random, then it needs an exponentially large sample and computational complexity to have a non-trivial fairness measure. The paper then showed that the convex hull of the occupancy measures induced by individual optimal policies is a good choice. Finally, the paper proposed a weighted RL objective that can be efficiently solved using traditional policy optimization algorithms to find a $q$-quantile optimal policy with respect to such an occupancy measure.

**Reviewers' Concerns**: Reviewers have appreciated the overall contributions. In particular, the reviewers appreciated the theoretical insights and how a policy optimization-based algorithm can be effective in solving fairness-related problems in MDP. Some of the reviewers have raised concerns as well. For example, reviewer tKms asked about the correct order of the number of calls for the optimization oracles. Reviewer Mguy raised questions on the motivation behind this work, the relationship with the other fairness measures, and the empirical validity. The authors have responded to the questions.

**AC's take**: While there was not much discussion between the reviewers and the authors, it seems that the authors' responses have addressed the questions well enough. The AC has also gone over the paper, and the AC agrees that the paper has made significant contributions that will be useful to the community. I would suggest that the authors to include the explanations in the main text. In particular, the  meaning of the policy aggregation and a detailed difference with the existing approaches would be super helpful.

**Reviewer Concerns:**

Reviewers have raised some concerns regarding this paper, as the reviewers received this paper with more positivity. According to the AC, the authors' responses have addressed them well.

**Reviewer Scores:**

Only reviewer Mguy has rated negatively. However, I believe that the authors' responses should alleviate the reviewer's concerns, and the reviewer would change their rating to be more positive.

---

### Decision · Program_Chairs · 2026-01-26

Accept (Poster)